# Studies on High-Resolution Airborne Synthetic Aperture Radar Image Formation with Pseudo-Random Agility of Interpulse Waveform Parameters

Zheng Ye *, Daiyin Zhu, Shilin Niu and Jiming Lv

College of Electronic and Information Engineering, Nanjing University of Aeronautics and Astronautics, Nanjing 211106, China; zhudy@nuaa.edu.cn (D.Z.); niushilin@nuaa.edu.cn (S.N.); jmlv_nj@nuaa.edu.cn (J.L.)
* Correspondence: yz1994@nuaa.edu.cn

**Abstract:** By means of alteration of the transmitted linear frequency modulation (LFM) signal waveform parameters, such as pulse width or chirp rate, initial phase, pulse repetition interval (PRI), and chirp rate polarity at every position of synthetic apertures, the pseudo-random agility technology of interpulse waveform parameters in airborne Synthetic Aperture Radar (SAR) actively increases the complexity and uncertainty of radar waveforms. This technology confuses jamming interception receivers, thus improving its anti-interference ability for active coherent jamming, which is one of the main research interests of airborne SAR technology. But the pseudo-random agility technology for interpulse waveform parameters faces certain challenges of large computation and complex system design, which need to be further studied and solved. To address these issues, a processing scheme of high-resolution SAR image formation which is appropriate for agile interpulse waveform parameters is proposed in this paper. This method can deal with multiple agile parameters, not only single ones as in most existing literature. Its computation load is nearly comparable to that of traditional SAR image formation with constant waveform parameters. The high-resolution SAR imaging results obtained by processing SAR raw data with agile interpulse waveform parameters demonstrate the effectiveness of the proposed method. In addition, real SAR images with resolutions of 0.5 m and 0.15 m, which are rarely found in the public literature, are shown under the circumstance of randomly changing the transmitted wideband LFM signal pulse parameters one by one.

**Keywords:** airborne Synthetic Aperture Radar (SAR); interpulse waveform parameters; pseudo-random agility; SAR electronic counter-countermeasures; active waveform countermeasure; active coherent interference; high-resolution imaging



## 1. Introduction

Conventional airborne SAR transmits wideband LFM signals with constant waveform parameters during the synthetic aperture data collection period, which are easy to be intercepted, sorted and identified by the jammer. Thus, it has the risk of interference, which reduces the reconnaissance efficiency of airborne SAR in electronic countermeasure environments [1]. Therefore, electronic countermeasures (ECMs) [1–4] have become the main challenge faced by airborne SAR sensors. To address this issue, electronic counter-countermeasures (ECCMs) [5–7] have also become one of the main research interests in the field of airborne SAR. Among various SAR ECM technologies, the most threatening one is active coherent interference technology [4,8]. It usually obtains accurate radar waveform parameters by interception and modulates the phase to repeat the radar signal or independently generates false target jamming signals which are coherent with the real radar echo signals, so that after this kind of jamming signals enter the radar, they can obtain high-enough coherent processing gain in range and azimuth dimensions. In this way, a good interference effect can be achieved with low jamming power, so it is an ideal SAR jamming method at present. The literature [8] points out that multi-channel SAR can

provide more spatial freedom, and it is a common active coherent interference suppression method [9]. But a multi-channel system will inevitably need more complex hardware and a heavier computation load, which is inconvenient for engineering implementation. The active waveform countermeasure method of single-channel SAR, which suppresses the active coherent interference by random agility of the interpulse waveform parameters, is a desirable, simple and effective anti-active coherent interference method. Generally, SAR transmits wideband LFM signals, and its waveform parameters include carrier frequency, bandwidth, pulse width, chirp rate, chirp rate polarity, initial phase and pulse repetition interval (PRI). While the bandwidth should be kept constant to ensure the equal range resolution of each pulse during the synthetic aperture data collection period, other parameters, such as pulse width or chirp rate [6], chirp rate polarity [10], initial phase [11] and PRI [12], can be set to random agility in principle. The random agility increases the complexity and uncertainty of the transmitted signal, confuses the jamming interception, disturbs the timing of coherent jamming repeater and destroys the coherence of the jamming signal in range and azimuth dimensions; finally, the purpose of suppressing coherent interference is achieved. The literature [13] points out that the pseudo-random agility technology for interpulse waveform parameters is faced with certain challenges of large computation and complex system design, which needs to be further studied and solved. Most of the existing works I the literature only study SAR anti-coherent interference technology [6,9–12] itself with random agility of interpulse waveform parameters. However, the influence of random agility of interpulse waveform parameters on airborne SAR image formation is ignored, and no definite conclusion, i.e., whether airborne SAR with random agility of interpulse waveform parameters can acquire high-quality high-resolution SAR images just like conventional airborne SAR with constant waveform parameters, was given. Although there are some works in the literature [12] that study SAR imaging based on non-uniform PRI, they mostly focus on a frequency domain reconstruction [14] for non-uniformly sampled signals in situations where the time offsets of each sampling instance are known to restore the uniform sampling, and then they carry out the normal imaging process. The validity of this reconstruction algorithm for non-uniformly sampled signals resulting from the random agility of interpulse PRI was not mentioned yet [12]. In addition, the process of transforming non-uniform azimuth sampling into the desired uniform azimuth sampling requires more computation and memory resources, which is also inconvenient for engineering implementation.

The purpose of this paper is to study a high-resolution airborne SAR image formation technique under the condition of pseudo-random agility of interpulse waveform parameters, and the main innovations are as follows.

(1) Considering the pseudo-random agility technology of interpulse waveform parameters in SAR, only a single parameter or two parameters, such as pulse width and/or initial phase, was changed in most previous works. The compensation for agile interpulse waveform parameters in SAR imaging processing is relatively easy, but its anti-interference ability for active coherent jamming is limited. The influence of agile interpulse waveform parameters, such as pulse width, chirp rate polarity, initial phase and PRI, on SAR imaging processing and their compensation methods are analyzed in detail. The computation load of the method proposed in this paper was comparable to that of traditional SAR image formation with constant waveform parameters. Obviously, the anti-interference ability for active coherent jamming should be improved through simultaneous random agility of pulse width, chirp rate polarity, initial phase and PRI.

(2) An innovatively designed and highly integrated Ku band 4.5 kg microminiature SAR sensor is briefly introduced, which consists of a co-aperture dual-polarized patch array antenna, two-axis gimbal, 80 W solid state power amplifier, dual-channel radio frequency (RF) receiver including a built in limiter, on-board processing board performing real-time image formation, 2 TB data storage board to store dual-channel raw data and real-time SAR images, necessary power conversion module and a

commercial off-the-shelf micro electro mechanical systems (MEMS)-based inertial measurement unit (IMU)/Global Position System (GPS) for motion measurement. Its performance specifications and several example SAR images with constant interpulse waveform parameters are shown to illustrate the proper function mode and excellent behavior of this 4.5 kg microminiature SAR sensor. Considerations to upgrade this SAR with the capability of the random agility of interpulse waveform parameters are given.

The rest of this paper is structured as follows. The influences of agile interpulse waveform parameters on SAR image formation are discussed in detail in Section 2; then, a flowchart of airborne SAR image formation is proposed. An innovative 4.5 kg microminiature SAR and the upgrading considerations on it with the capability of random agility of interpulse waveform parameters are introduced briefly in Section 3. Experimental verification results of SAR imaging, which were obtained through processing this upgraded SAR raw data according to the proposed flowchart of airborne SAR image formation under the circumstance of randomly changing the transmitted wideband LFM signal waveform pulse parameters one by one, are shown in Section 4. Section 5 gives some discussion. Section 6 summarizes the whole paper.

## 2. Influences of Agile Interpulse Waveform Parameters on SAR Image Formation

At every pulse repetition frequency (PRF) moment, SAR transmits wideband LFM signals, whose waveform parameters, such as carrier frequency, pulse width or chirp rate, polarity, initial phase and PRI, can be randomly altered within a certain range, except that the bandwidth should be kept constant. The impacts of these variations in waveform parameters on the imaging process are analyzed as follows.

### 2.1. Agility of Carrier Frequency

If we change carrier frequency randomly within the tolerable range of radar hardware at every PRF moment, the anti-jamming ability of the radar can be improved in principle. Assuming that the nominal carrier frequency of the radar is $f_c$, $f_c^i$ denotes the carrier frequency of the $i$th pulse and can be expressed as

$$f_c^i = f_c + \Delta f^i \tag{1}$$

where $\Delta f^i$ is the random frequency offset of the $i$th pulse from the nominal carrier frequency.

The spotlight SAR data collection geometry is illustrated in Figure 1. Without the loss of generality, assume that the radar undergoes an arbitrary flight path. The instantaneous position of the radar antenna phase center (APC) is denoted by the vector $\vec{r}_o(t) = [x_a(t), y_a(t), z_a(t)]$ at slow time $t$. The variables $\theta = \theta(t)$ and $\varphi = \varphi(t)$ represent the instantaneous azimuth angle and grazing angle, respectively. Denote $\theta_{ref}$ as the reference azimuth angle and $\varphi_{ref}$ as the corresponding grazing angle at slow time $t = 0$. Consider a ground point target scatterer, whose coordinates are defined by the vector $\vec{r}_c = (x_c, y_c, 0)$. The vector between the APC and this target scatterer is $\vec{r}(t) = \vec{r}_o(t) - \vec{r}_c$.

In the case of constant carrier frequency $f_c$, and ignoring a minor phase term often referred to as the residual video phase error (RVPE), the 2-D phase history [15,16] delivered to the Polar Format Algorithm (PFA) after motion compensation with respect to the scene center is expressed as

$$S_1(t, f_\tau) = C \cdot \exp\left\{ j \frac{4\pi(f_c + f_\tau)}{c} \left[ \left| \vec{r}_o(t) \right| - \left| \vec{r}(t) \right| \right] \right\} \tag{2}$$

where $t$ represents slow time, $f_\tau$ is range frequency, $c$ is the speed of propagation, and $C$ includes the nonessential factors of the transmitted pulse envelop and azimuth antenna pattern.

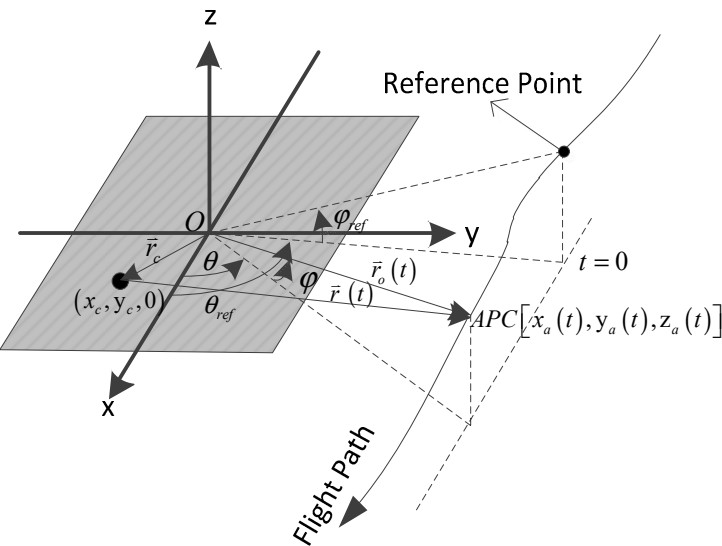

**Figure 1.** Spotlight SAR data collection geometry.

Similarly, the 2-D phase history when the radar carrier frequency is agile can be expressed as

$$
\begin{aligned}
S_2(t, f_\tau) &= C \cdot \exp\left\{ j\frac{4\pi\left(f_c^i + f_\tau\right)}{c}\left[\left|\vec{r}_o(t)\right| - \left|\vec{r}(t)\right|\right]\right\} \\
&= C \cdot \exp\left\{ j\frac{4\pi\left(f_c + f_\tau\right)}{c}\left[\left|\vec{r}_o(t)\right| - \left|\vec{r}(t)\right|\right]\right\} \exp\left\{ j\frac{4\pi\cdot\Delta f^i}{c}\left[\left|\vec{r}_o(t)\right| - \left|\vec{r}(t)\right|\right]\right\}
\end{aligned}
\tag{3}
$$

Comparing Equation (2) with Equation (3), we can see that the first phase term in the second line of Equation (3) is the same as Equation (2), which is the basic imaging term containing the useful information about locating and focusing the scene target. But the second phase term resulting from the random frequency offset $\Delta f^i$ is undesirable and needs to be eliminated to restore the correct 2-D phase history.

At every synthetic aperture time instant of $t$, the random frequency offset $\Delta f^i$ and the instantaneous distance $\left|\vec{r}_o(t)\right|$ are fixed values, and the instantaneous distance $\left|\vec{r}(t)\right|$ is a variable related to the position of every target scatterer in the imaging scene, so the second phase term is space-variant. Obviously, the compensation of the second phase term to restore the correct 2-D phase history requires a heavy computation load, which makes it impractical for engineering implementation. So, we do not recommend changing carrier frequency randomly.

### 2.2. Agility of Pulse Width and Chirp Rate Polarity

Assume that the bandwidth $B$ of the transmitted LFM signal is kept constant. $a^i$ denotes the chirp rate polarity, which is randomly +1 or $-1$, and the superscript $i$ represents the $i$th pulse. Bandwidth is expressed as

$$
B = K_r^i \cdot \tau_p^i
\tag{4}
$$

where $K_r^i$ and $\tau_p^i$ denote the chirp rate and pulse width, respectively. So, the range match filter $H$ is

$$
H_r^i(f_\tau) = \exp\left( j \cdot a^i \cdot \frac{\pi}{K_r^i} \cdot f_\tau^2 \right)
\tag{5}
$$

where $f_\tau$ is the range frequency. As can be seen from Equation (5), the range match filter needs to be changed pulse by pulse according to the current chirp rate and chirp rate polarity. Obviously, this is different from the traditional range match filter with constant waveform parameters.

The random agility of the pulse width and chirp rate polarity will not affect the maximum range processing gain obtained by pulse compression of the echo signal, while the jamming signal cannot obtain sufficient range processing gain due to chirp rate or chirp rate polarity mismatch; active coherent interference can be suppressed effectively in this way [6].

Because the processing gain [17] of the range match filter is $\tau_p^i \cdot B$, the agility of the pulse width will lead to fluctuation in the processing gain, resulting in non-uniform image intensity from pulse to pulse. Moreover, the non-constant pulse width $\tau_p^i$ and $PRI^i$ will lead to a variable duty factor, which is represented as

$$d^i = \frac{\tau_p^i}{\mathrm{PRI}^i} \tag{6}$$

and it means the interpulse fluctuation of average power $P_{av}^i$ transmitted during the synthetic aperture data collection period. $P_{av}^i$ is expressed as

$$P_{av}^i = P_t \cdot d^i \tag{7}$$

where $P_t$ represents the peak transmitter power and is usually a constant value, which will also lead to non-uniform image intensity from pulse to pulse. Therefore, the random agility of the pulse width or PRI will eventually affect the uniformity of the SAR image, which needs to be compensated for.

Assume that the pulse width $\tau_p^0$ and pulse repetition interval $PRI^0$ were chosen according to the requirements of the operating range and Signal to Noise Ratio (SNR) of the airborne SAR with constant waveform parameters. The processing gain equalization factor and the power equalization factor with agile waveform parameters are, respectively, expressed as

$$A_1^i = \frac{\tau_p^i \cdot B}{\tau_p^0 \cdot B} = \frac{\tau_p^i}{\tau_p^0} \tag{8}$$

$$A_2^i = \frac{d^i}{d^0} = \frac{\tau_p^i \cdot \mathrm{PRI}^0}{\tau_p^0 \cdot \mathrm{PRI}^i} \tag{9}$$

Therefore, the equalization of the processing gain and average power fluctuations resulting from the random agility of pulse width and PRI can be achieved by multiplying the amplitude of the $i$th echo signal by a factor $\frac{1}{\sqrt{A_1^i A_2^i}}$.

### 2.3. Agility of Initial Phase

Airborne SAR can randomly change the initial phase $\phi_n^i$ of the transmitted wideband LFM signal on purpose at every PRF moment. $\phi_n^i$ represents the initial phase of the $i$th pulse, which is a uniformly distributed random number between $[0, 2\pi]$ and is statistically independent from pulse to pulse. After receiving the echo signal, the random initial phase can be compensated for by multiplying the echo signal by a phase $\exp(-j\phi_n^i)$ according to the prior information of the initial phase of the $i$th pulse, so as to restore the coherence of the azimuth signal during the synthetic aperture data collection period. The subsequent azimuth processing is the same as that of conventional SAR. This random initial phase will cause the azimuth phase of the jamming signal to be not perfectly coherent, and the azimuth processing gain cannot be obtained sufficiently, so that the jamming signal can be suppressed to a certain extent [11].

### 2.4. Agility of PRI

The agility of PRI makes it difficult for the jammer to apply fully coherent jamming signal with airborne SAR, resulting in a certain degree of ineffectiveness of the coherent jamming signal. However, for airborne SAR itself, the random agility of PRI causes random

fluctuation in the duty factor, resulting in variation in average power, which leads to non-uniform image intensity from pulse to pulse. A compensation technique was given in Section 2.2. Moreover, non-uniform azimuth sampling caused by the random agility of PRI also leads to a poor azimuth sidelobe if the image formation process is not appropriate for non-uniform azimuth sampling, and neglecting this non-uniformity results in the deterioration of SAR image quality [12].

According to Ref. [15], the 2-D phase history in Equation (2) after range resampling in PFA is

$$S_3(t, f_\tau) = C \cdot \exp\left\{ j\frac{4\pi(f_c + f_\tau)}{c} \cos\varphi_{ref} \cdot (x_c \cot\theta + y_c) \right\} \tag{10}$$

where $\cot\theta$ is a function of $t$. As analyzed in Section 2.3 of Ref. [15], even for constant PRI, which means uniformly spaced time instants of $t$, the corresponding values of $\cot\theta$ are, in general, non-uniformly spaced for the realistic motion of the APC. However, the range-resampled phase history in Equation (10) can be interpolated on specific time instants to achieve uniformly spaced values for the term $\cot\theta$, which is a linear function of $t$.

Finally, after azimuth resampling [15], the 2-D phase history becomes

$$S_4(t, f_\tau) = C \cdot \exp\left[ j\frac{4\pi}{c} \cos\varphi_{ref} \cdot (\Omega \cdot x_c f_c t + y_c f_\tau) \right] \exp\left( j\frac{4\pi f_c}{c} \cos\varphi_{ref} y_c \right) \tag{11}$$

where $\Omega$ is a constant value [15] determined by the azimuth interpolation process.

In Equation (11), there has been no cross-product quadratic phase term of $t$ and $f_\tau$. The 2-D sinusoid in Equation (11) can be converted into a focused target response via the 2-D Fourier transform with respect to $t$ and $f_\tau$.

Obviously, the azimuth resampling procedure in PFA leads to azimuth sample spacing equalization in SAR data processing. So, the PFA is naturally suitable for non-uniform azimuth sampling caused by the random agility of PRI. The smooth embedment of the equalization into the image formation algorithm is a unique advantage that the PFA has over the other precise SAR image formation algorithms, such as the Range Migration Algorithm (RMA) [18–20].

*2.5. Airborne SAR Image Formation*

There are several typical and commonly used SAR image formation algorithms, such as the Range Doppler Algorithm (RDA) [18,19], Chirp Scaling Algorithm (CSA) [18–21], RMA [22] and PFA [23–28].

Considering that PFA performs phase adjustment and compensation in the time domain, it is not only simple, efficient and easy to implement in engineering, but it can also effectively compensate for the non-ideal trajectory of a radar platform and is naturally suitable for non-uniform azimuth sampling caused by the random agility of PRI [15]. Thus, the time-consuming processing step for transforming non-uniform azimuth sampling into the desired uniform azimuth sampling [14] can be omitted. However, PFA has the space-variant phase error caused by planar wavefront assumption [23–25], which limits the scene size of SAR images without defocus effects and causes the obvious geometric distortion of SAR images. Space-Variant Post-Filtering (SVPF) [16,25–28] is adopted to correct wavefront curvature and thus to enlarge the scene size limits and eliminate geometric distortion in polar-formatted SAR imagery. Therefore, we propose to use PFA + SVPF + PGA [29–31] as a processing scheme of high-resolution SAR image formation, which is appropriate for agile interpulse waveform parameters.

The proposed imaging process flowchart is shown in Figure 2, in which random initial phase and pulse-to-pulse amplitude fluctuation that are caused by variable pulse width and PRI are compensated for during preprocessing, as shown in the blue dotted line box. Pulse compression and rotating platform compensation are shown in the green box, which completes the range match filter, as shown in Equation (5), and stripmap-to-spotlight conversion. It should be noted that the range match filter needs to be changed pulse by pulse according to the interpulse agility of pulse width and chirp rate polarity, which is

different from traditional pulse compression with constant waveform parameters. The standard polar format processing flowchart [32] is shown in the blue box, and SVPF plus PGA are shown in the yellow box to accomplish wavefront curvature correction [16,28] and phase gradient autofocus [33]. It can be seen that compared with conventional airborne SAR processing, only the random initial phase and amplitude compensation in the blue dotted line box are added as a preprocessing step, and the range match filter in the green box needs to be calculated pulse by pulse, so its resulting computation can be neglected.

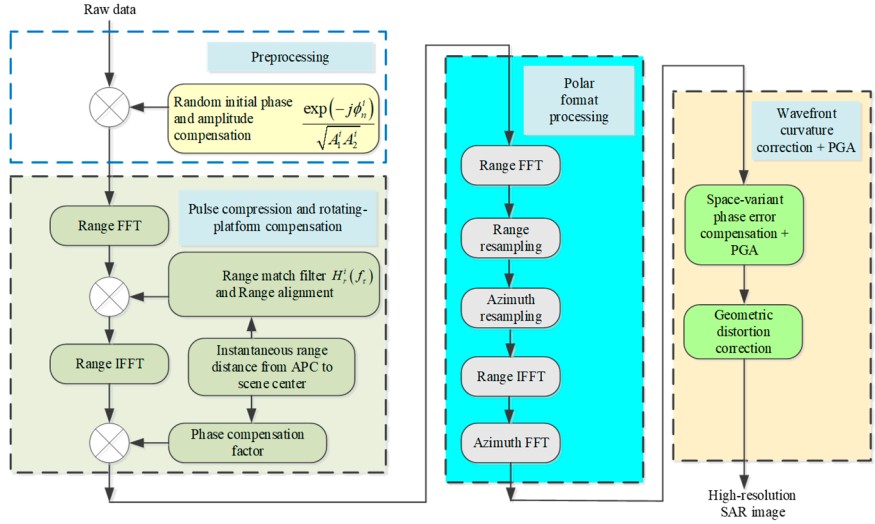

**Figure 2.** Flowchart of airborne SAR image formation with agile interpulse waveform parameters.

## 3. Considerations on Upgrading A SAR with Agile Interpulse Waveform Parameters

### 3.1. Brief Introduction of 4.5 kg Microminiature SAR

A Ku band microminiature SAR is chosen to improve the capability of pseudo-random agile interpulse waveform parameters. This SAR was innovatively designed and integrated all of the following function modules into a single unit, as shown in Figure 3: an antenna/gimbal, transmitter/receiver, processing/data storage board, power conversion module and MEMS based IMU/GPS. Figure 3a is a physical photo, Figure 3b is the front view, Figure 3c is the rear view and Figure 3d is the assembly view of this microminiature SAR. In addition, the inclusion of a host computer and display (either onboard or via a remote link) completes this SAR system. Table 1 outlines the top-level performance specifications for this SAR system.

**Table 1.** 4.5 kg microminiature SAR specifications.

| Parameter | | Value | Comments |
|---|---|---|---|
| Weight | | 4.3 kg | measured value |
| Size (length × width × height) | | 324.7 mm × 98 mm × 164.5 mm | measured value |
| Power | | +28 VDC at 125 W max | measured value |
| Frequency Band | | Ku | |
| RF Bandwidth | | 1500 MHz max | |
| Co-aperture dual-polarized patch array Antenna | Aperture size | 250 mm × 90 mm | Measured value at center frequency. |
| | Gain | 21.5 dB | |
| | Beamwidth | 8.1° AZ/19.6° EL | |
| | Sidelobe | −20.5 dB | |
| | Isolation | −35 dB | |

**Table 1.** *Cont.*

| Parameter | | Value | Comments |
|---|---|---|---|
| Gimbal Type | | Azimuth and Elevation Dual Axis | Azimuth coverage: $-90° \pm 30°$ or $+90° \pm 30°$, Elevation coverage: 0~180°. |
| Transmitter/Receiver with Exciter module | Transmitter Power | 80 W peak | Solid state power amplifier |
| | Noise Figure of RF Receiver | 4 dB | Includes built in limiter and dual channel receiver. |
| RF losses | | ~3 dB | Transmitter output to LNA/Limiter input. |
| Processing/data storage module | Processing board | 1 FPGA + 2 multi-core DSP | On-board real-time image formation |
| | Data storage board | 2 TB SSD | Store dual-channel raw data and real-time SAR images |
| Power conversion module | | Output all kinds of low-voltage power supply required by each module and cooling fan | The input is 28 VDC. |
| Motion sensor | | MEMS based IMU/GPS | COTS (commercial off-the-shelf) |
| SAR Imaging modes | | Spotlight and stripmap | |
| Resolution | Spotlight | 0.15 m, 0.3 m, 0.5 m optional | |
| | Stripmap | 0.3 m, 0.5 m, 1 m, 3 m optional | |
| Maximum Range | | ≮15 km | Assume 3 m resolution, $-25$ dB noise equivalent reflectivity |
| Image Size | | 8 K × 8 K pixels max | Both along-track and cross-track for spotlight mode |
| SAR Imaging formation on-board | | CS-PFA + SVPF + PGA | Space-Variant Post-Filtering [16,28] based Polar Format Algorithm using Chirp Scaling [32] with Phase Gradient Autofocus [33] |

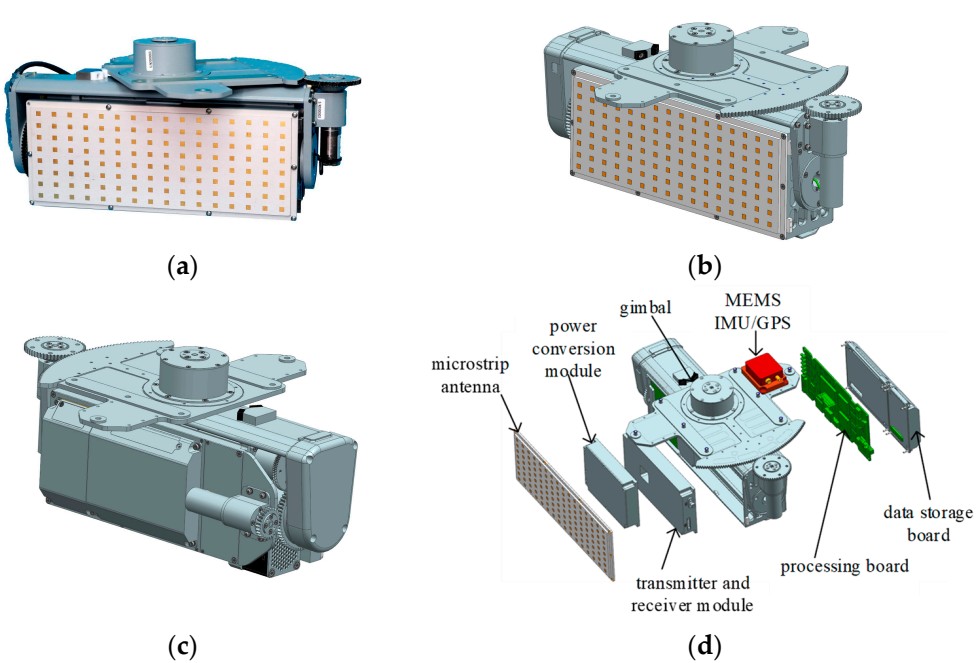

**Figure 3.** 4.5 kg microminiature SAR: (**a**) physical photo; (**b**) front view; (**c**) rear view; (**d**) assembly view.

This Ku band 4.5 kg microminiature SAR has been flight-demonstrated on a variety of manned aircraft, such as the Y12, Cessna 208 B and Robinson R44, and small unmanned aerial vehicles (SUAVs), such as fixed-wing or multi-rotor UAVs. A large amount of high-resolution stripmap and spotlight dual-polarization SAR raw data and images were acquired. Here, we show just a few examples of the data product, illustrating the proper function mode and excellent performance of this SAR sensor with constant waveform parameters.

The 0.5 m × 0.5 m resolution dual-polarization stripmap SAR images are shown in Figure 4, which were obtained in the first flight demonstration of this 4.5 kg microminiature SAR in September 2020 on a Y12. From the rural area, we can see the obvious RCS difference between the VV-polarized SAR image shown in Figure 4a and the VH-polarized SAR image shown in Figure 4b.

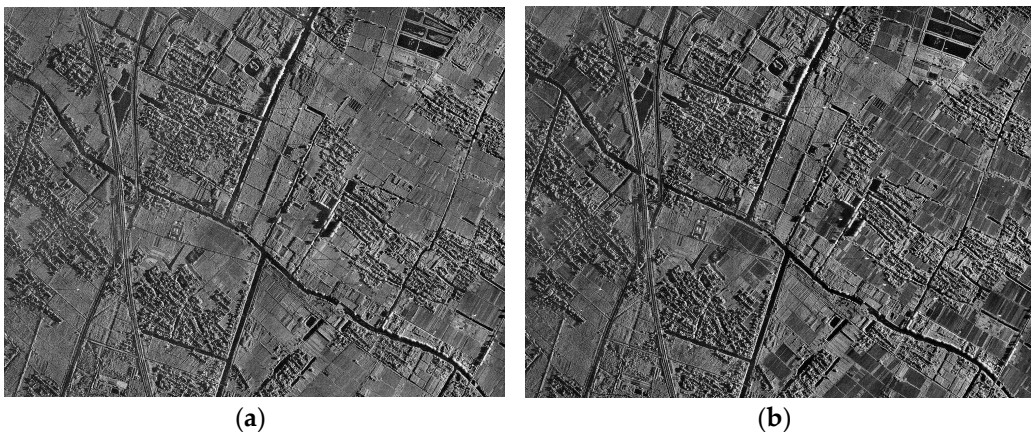

(**a**)         (**b**)

**Figure 4.** Dual-polarization stripmap SAR image, 0.5 m resolution: (**a**) VV polarization; (**b**) VH polarization.

Next, the real-time stripmap SAR image of a rural area at 0.3 m × 0.3 m resolution generated by this radar sensor is shown in Figure 5, which was flight tested on a small fixed-wing UAV.

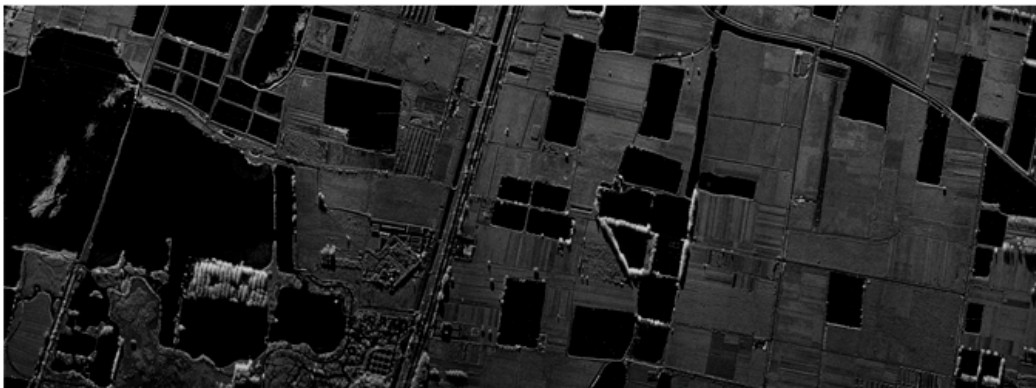

**Figure 5.** Real-time stripmap image of rural area at 0.3 m resolution.

Finally, four real-time spotlight SAR images of an urban area at 0.15 m × 0.15 m resolution acquired from different view angles of the same track are shown in Figure 6, whose scene center is the ground parking lot of a hotel. Their corresponding azimuth angle is 50°, 53.71°, 57.81° and 62.31°, respectively.

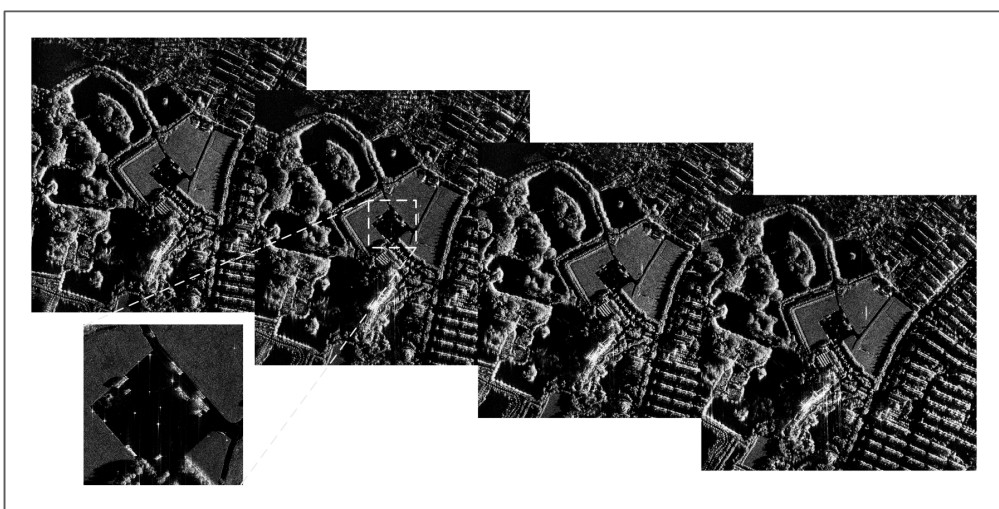

**Figure 6.** Real-time spotlight images of urban area at 0.15 m resolution.

It can be seen that the SAR images of different resolutions shown above from Figures 4–6 are well focused, and they have clear and rich texture and uniform intensity under the condition of constant waveform parameters.

### 3.2. Upgrading the 4.5 kg Microminiature SAR with Agile Interpulse Waveform Parameters

Considering its wideband LFM signal is generated by a digital waveform synthesizer, it is convenient to upgrade this radar with the capability of pseudo-random agility of interpulse waveform parameters such as pulse width, chirp rate polarity, initial phase and PRI at every PRF moment. A 0.5 m and 0.15 m resolution mode was chosen to be upgraded. Table 2 is the original waveform parameters of 0.5 m and 0.15 m resolution mode.

**Table 2.** Radar original waveform parameters.

| Waveform Parameters | Value |
|---|---|
| Resolution mode/m | 0.5/0.15 |
| Bandwidth/MHz | 400/1200 |
| Chirp rate polarity | + |
| Pulse width/μs | 10 |
| PRI/μs | 1000 |

When the radar waveform parameters are agile, in order to ensure that the duty factor change caused by variable pulse width or PRI do not cause too much change in radar range distance or SNR, it is necessary to limit the variation range of the pulse width and PRI. According to the radar equation [17,22] of SAR, $R^3 \propto P_{av} \propto d$, if the variation of radar range is required to be less than 10%, then the estimated allowable variation range of pulse width and PRI is about 9~11 μs and 900~1100 μs, respectively. Therefore, we have determined the random agility mode for pulse width, PRI, initial phase and chirp rate polarity. Table 3 is a description of four agile parameters in our experiment.

**Table 3.** The description of four agile parameters.

| Agile Parameters | Variation Range | Step Width |
|---|---|---|
| Pulse width/μs | 9~11 | 80 ns |
| PRI/μs | 900~1100 | 80 ns |
| Initial phase/° | 0~360° | $360°/2^{14}$ |
| Chirp rate polarity | −1 or +1 | |

## 4. Experimental Verification of SAR Image Formation with Agile Interpulse Waveform Parameters

The 4.5 kg microminiature SAR was upgraded according to the considerations introduced in Section 3.2. Owing to no suitable jammer at hand, and given that the main purpose of this work is to study and verify whether airborne SAR with agile interpulse waveform parameters can acquire high-quality high-resolution SAR images just like conventional airborne SAR with constant waveform parameters, this upgraded 4.5 kg SAR was flight tested independently, and SAR raw data with resolutions of 0.5 m and 0.15 m were collected with agile interpulse waveform parameters.

The variations in the agile pulse width, PRI, initial phase and chirp rate polarity of the first 1000 pulses in one frame of raw data collected by the upgraded 4.5 kg SAR with a resolution of 0.5 m are shown in Figure 7. It can be shown that four waveform parameters, including pulse width, PRI, initial phase and chirp rate polarity, are agile simultaneously at every PRF moment; therefore, the complexity and uncertainty of radar waveform can be greatly increased, and the anti-interference ability for active coherent jamming can be significantly improved in principle.

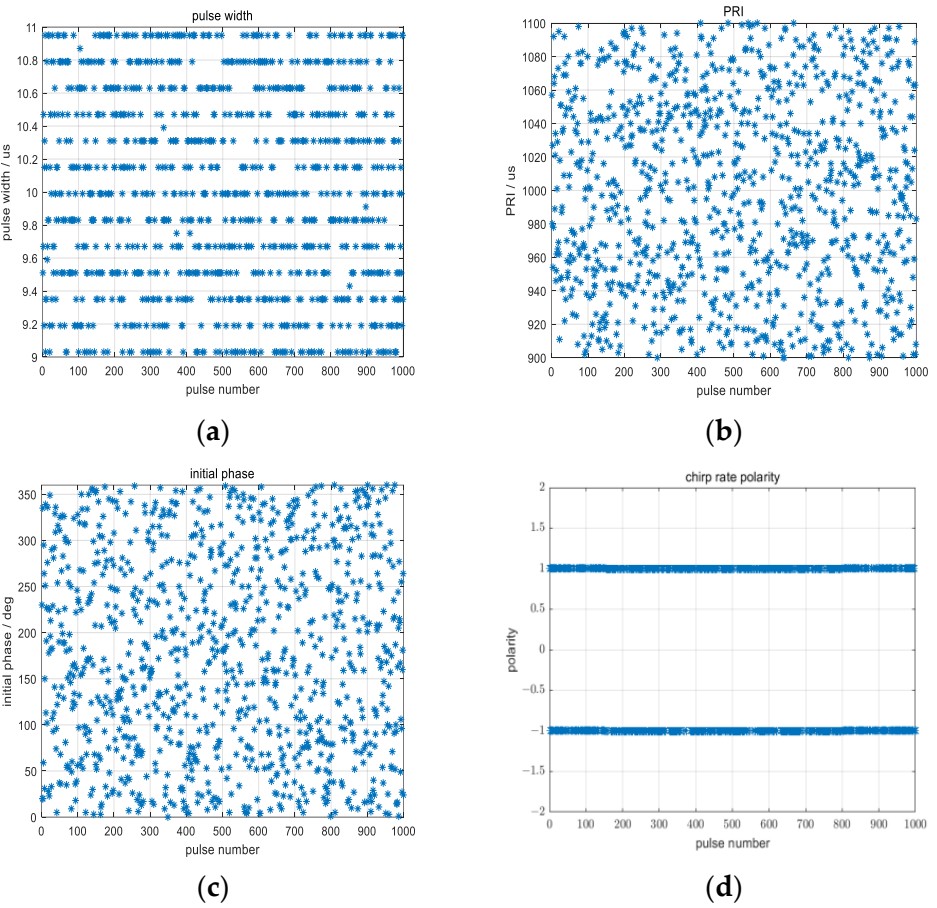

**Figure 7.** Agile interpulse waveform parameters at resolution 0.5 m: (**a**) pulse width; (**b**) PRI; (**c**) initial phase; (**d**) chirp rate polarity.

The amplitude equalization factor required for compensating the fluctuations in processing gain and average power resulting from the random agility of pulse width and PRI is shown in Figure 8. It can be found that the range of the amplitude equalization factor is about 0.87 to 1.17.

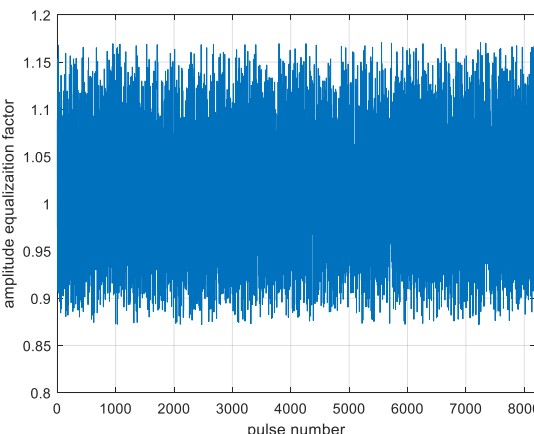

**Figure 8.** Amplitude equalization factor for compensating for the fluctuations in processing gain and average power resulting from the random agility of pulse width and PRI at a resolution of 0.5 m.

The 0.5 m resolution rural area SAR images are shown in Figure 9. Figure 9a is a small area from the imaging result obtained according to the processing flowchart shown in Figure 2, except that the agility of the pulse width or chirp rate is not compensated for, which corresponds to the white dotted line box in the SAR image given in Figure 9f. Similarly, Figure 9b–d are the same area when the agility of chirp rate polarity, initial phase and PRI are also not compensated for, respectively. It can be seen that Figure 9a is severely defocused, which is due to the mismatch of chirp rate resulting from pulse width agility, and Figure 9b–d have various degrees of SNR deterioration. The SNR deterioration in Figure 9c is due to the incoherence in the azimuth caused by initial phase agility, and the image deterioration in Figure 9d is due to poor sidelobe in the azimuth caused by PRI agility. The reason for the image deterioration shown in Figure 9b will be explained in the following section. Figure 9e shows the rural area SAR image obtained by processing the SAR raw data with no compensation for agile interpulse waveform parameters; obviously, it is completely defocused. Figure 9f shows the rural area SAR image obtained by processing the same SAR raw data with compensation for agile interpulse waveform parameters according to the proposed processing flowchart shown in Figure 2. It can be seen that the SAR image shown in Figure 9f is well focused and has clear and rich texture and uniform intensity.

Due to the random agility of chirp rate polarity, when the chirp rate polarity of the jamming signal matches the transmitted signal, the jamming signal can normally obtain the range processing gain, but when it is mismatched, the jamming signal cannot obtain the processing gain. The result of pulse compression and rotating-platform processing with no compensation for chirp rate polarity agility is shown in Figure 10a, and the result of pulse compression and rotating-platform processing with compensation for chirp rate polarity agility is shown in Figure 10c. Figure 10b,d are the local enlarged results of their corresponding area marked by yellow dotted line boxes. Comparing the vertical bright line in Figure 10b with that in Figure 10d, we can see that the vertical lines are intermittent and continuous, respectively. It can be found that the processing result as indicated by Figure 10a resulted from a chirp rate polarity mismatch and is equivalent to the random loss of samples or unequally spaced sampling in the azimuth dimension, so after normal azimuth processing, the image will be deteriorated by azimuth mainlobe broadening and a poor sidelobe, just like the effect shown in Figure 9b.

The contrast value adopted in the objective evaluation is as follows:

$$C = \frac{\sigma[|I(m,n)|]}{E[|I(m,n)|]} = \frac{\sqrt{E\left\{[|I(m,n)| - E(|I(m,n)|)]^2\right\}}}{E[|I(m,n)|]} \tag{12}$$

where $\sigma(\cdot)$ represents the standard deviation of the image intensity, $E(\cdot)$ represents the mean of the image intensity and $|I(m,n)|$ represents the intensity of the pixel at $(m, n)$ in the image. The higher the contrast value, the higher the image quality.

The contrast values corresponding to Figure 9a–d and the same area indicated by the white dotted line box in Figure 9f are presented in Table 4. It can be seen that the area indicated by the white dotted line box in Figure 9f has the highest contrast value, which is at least twice higher than that of Figure 9a–d.

**Table 4.** Contrast values corresponding to Figure 9a–d and the same area indicated by the white dotted line box in Figure 9f.

|  | Figure 9a | Figure 9b | Figure 9c | Figure 9d | Figure 9f |
|---|---|---|---|---|---|
| Contrast value | 1.005 | 10.602 | 5.045 | 14.946 | 30.859 |

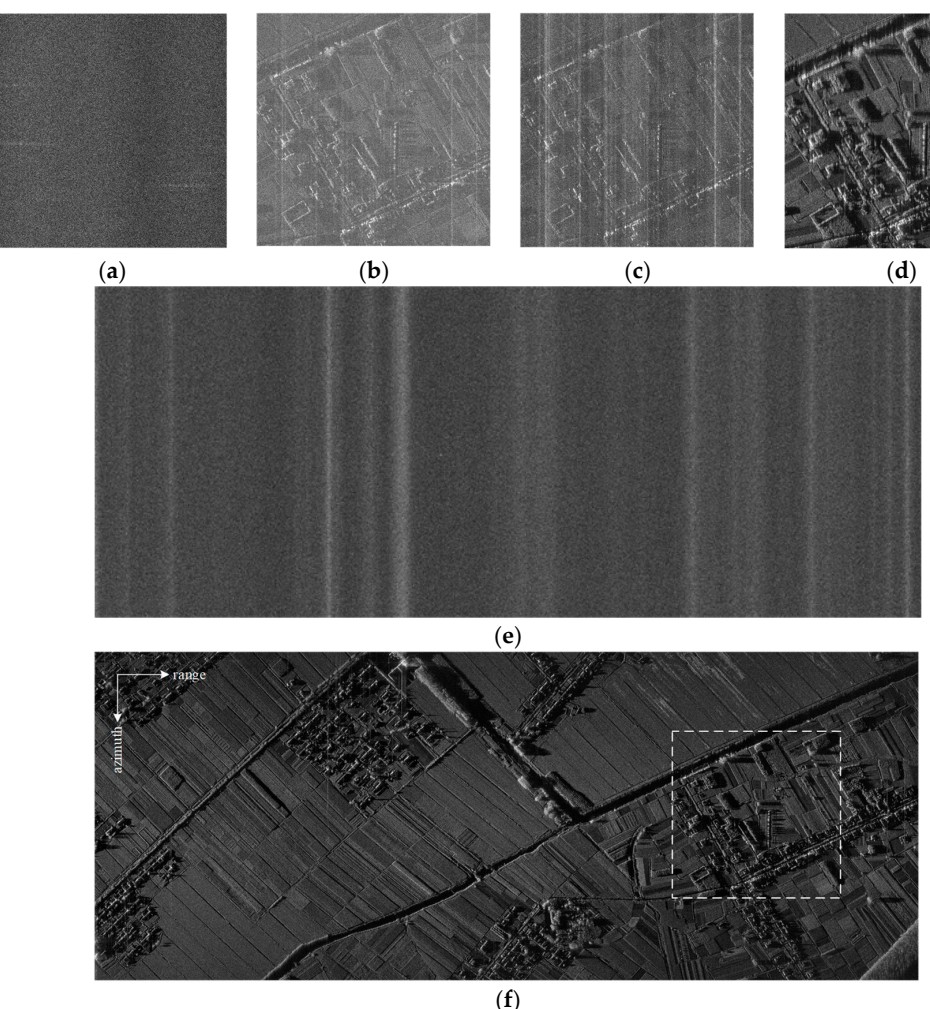

**Figure 9.** Rural area SAR images at 0.5 m resolution with agile interpulse waveform parameters: (**a**) no compensation for agility of pulse width or chirp rate; (**b**) no compensation for agility of chirp rate polarity; (**c**) no compensation for agility of initial phase; (**d**) no compensation for agility of PRI; (**e**) no compensation for all agile interpulse waveform parameters; (**f**) compensation for all agile interpulse waveform parameters according to the flowchart shown in Figure 2.

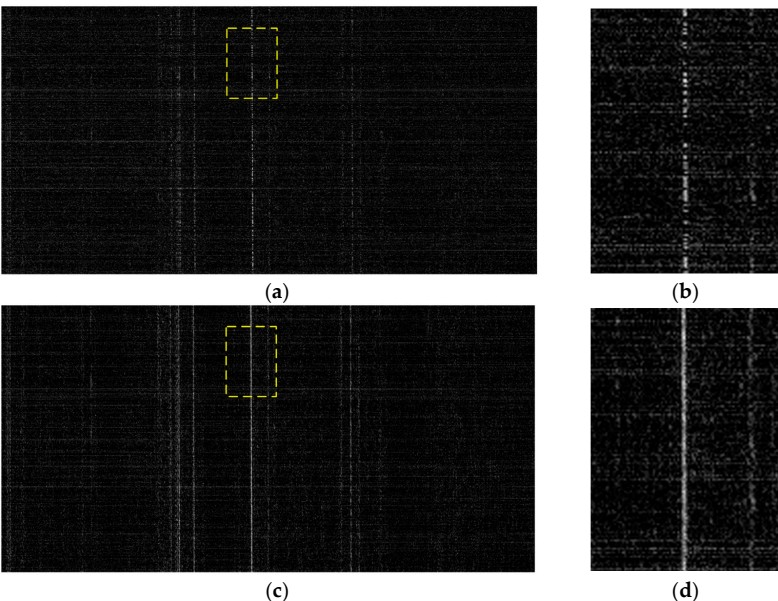

**Figure 10.** The comparison of the chirp rate polarity of the jamming signal matches and mismatches the transmitted signal: (**a**) result of pulse compression and rotating-platform processing with no compensation of chirp rate polarity agility; (**b**) local enlarged result of the area marked by yellow dotted line box of (**a**); (**c**) result of pulse compression and rotating-platform processing with compensation of chirp rate polarity agility; (**d**) local enlarged result of the area marked by yellow dotted line box of (**c**).

The variations in the agile pulse width, PRI, initial phase and chirp rate polarity of the first 1000 pulses in one frame of raw data collected by the upgraded 4.5 kg SAR with resolution 0.15 m are shown in Figure 11.

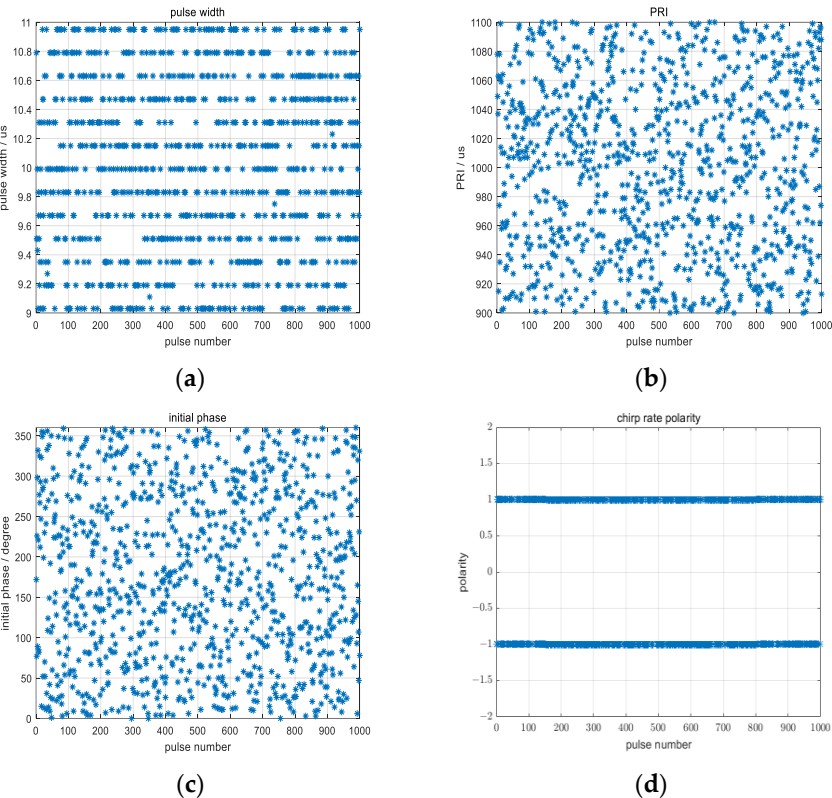

**Figure 11.** Agile interpulse waveform parameters at resolution 0.15 m: (**a**) pulse width; (**b**) PRI; (**c**) initial phase; (**d**) chirp rate polarity.

The amplitude equalization factor required for compensating the fluctuations in processing gain and average power resulting from the random agility of pulse width and PRI is shown in Figure 12. As can be seen from Figure 12, the range of the amplitude equalization factor is also about 0.87 to 1.17.

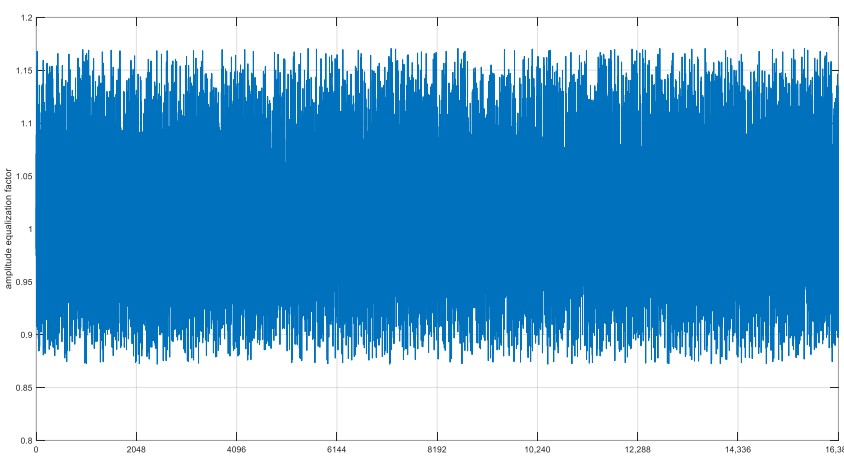

**Figure 12.** Amplitude equalization factor for compensating the fluctuations of processing gain and average power resulted from the random agility of pulse width and PRI at a resolution of 0.15 m.

According to the same processing flow shown in Figure 2, not surprisingly, the same high-quality SAR image with 0.15 m resolution can be obtained. Figure 13 shows a rural area scenario, with the power transmission line, the pylon reflection, its shadow and power transmission line shadow clearly visible in the spotlight SAR image at a resolution 0.15 m with agile interpulse waveform parameters.

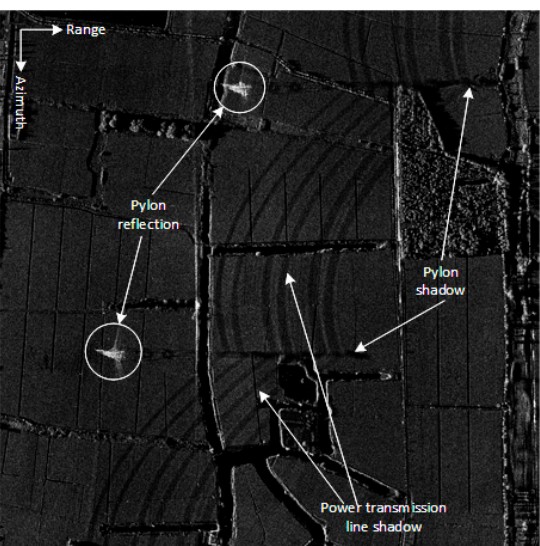

**Figure 13.** Spotlight SAR image of a power transmission line scene at 0.15 m resolution with agile interpulse waveform parameters.

The high-resolution SAR imaging results shown in Figures 9f and 13, and the contrast values shown in Table 4, indicate that airborne SAR can acquire the same satisfactory high-quality SAR images definitely under the condition of random agility of interpulse waveform parameters, just as the images shown in Figures 4–6 with constant waveform parameters.

## 5. Discussion

In most previous works, the active waveform countermeasure method of SAR generally uses only single agile parameter or two agile parameters such as pulse width and/or initial phase, which is relatively easy for engineering implementation, but with limited anti-active coherent interference ability. The simultaneous random agility of four waveform parameters of SAR, namely pulse width, chirp rate polarity, initial phase and PRI, was proposed and verified in this paper; therefore, the complexity and uncertainty of the radar waveform were greatly increased, and the anti-interference ability can certainly be improved in principle.

Based on the detailed analysis of the 2-D phase history of SAR as illustrated in Equation (2), we find that random agility of carrier frequency is not recommended due to its compensation requiring a heavy computation load. And this finding has rarely been seen in previous studies in the literature.

Comparing the results of experimental verification of SAR image formation with agile interpulse waveform parameters, as shown in Figures 9f and 13, with the example images of Figures 4–6, acquired by the 4.5 kg microminiature SAR sensor with constant waveform parameters, it can be seen that airborne SAR can acquire satisfactory high-quality SAR images definitely under the condition of random agility of interpulse waveform parameters through our proposed processing flowchart shown in Figure 2.

We had focused on the study of high-resolution airborne SAR image formation under the condition of pseudo-random agility of interpulse waveform parameters in this paper. In the future, we will try to carry out practical countermeasures against active coherent jammer-type systems to verify the actual suppression effect of our proposed method.

## 6. Conclusions

The pseudo-random agility technology of interpulse waveform parameters can certainly improve the anti-active coherent interference ability of airborne SAR. However, this pseudo-random agility technology itself will bring some difficulties and challenges to accurate SAR imaging. The main purpose of this work is to study and verify whether airborne SAR with agile interpulse waveform parameters can acquire high-quality high-resolution SAR images just like conventional airborne SAR with constant waveform parameters. The main findings of this paper are as follows.

(1) The influences of agile interpulse waveform parameters on SAR imaging were analyzed. It can be seen that pulse width or chirp rate, polarity, initial phase and PRI can be randomly altered within a certain range and also can be easily compensated during the SAR imaging process. But, changing carrier frequency randomly is not recommended due to its compensation requiring a heavy computation load. From the above analysis, an efficient processing flowchart of high-resolution SAR image formation which was appropriate for agile interpulse waveform parameters was proposed. Its computation load was comparable to that of traditional SAR image formation with constant waveform parameters.

(2) A Ku band 4.5 kg microminiature SAR, which was innovatively designed and highly integrated, was introduced briefly. Several example SAR images with constant interpulse waveform parameters were shown. Considerations to upgrade this SAR with the capability of random agility of interpulse waveform parameters were given.

(3) SAR raw data with agile interpulse waveform parameters were acquired using this upgraded SAR sensor. Then, real high-resolution SAR images with resolutions of 0.5 m and 0.15 m were processed successfully from the acquired SAR raw data and were shown under the circumstance of randomly changing the transmitted wideband LFM signal waveform pulse parameters one by one.

The above research results fully indicate that the proposed processing scheme of high-resolution SAR image formation for agile interpulse waveform parameters studied in this paper is simple and effective, which lays a solid foundation for future fielding airborne SAR

to counter active coherent interference in complex electromagnetic environments by means of active waveform countermeasure, and it has been of great practical value in engineering.

**Author Contributions:** Conceptualization, Z.Y. and D.Z.; methodology, Z.Y.; software, Z.Y.; validation, Z.Y.; formal analysis, S.N.; investigation, S.N.; data curation, Z.Y.; writing—original draft preparation, Z.Y.; writing—review and editing, Z.Y.; supervision, J.L.; project administration, D.Z.; funding acquisition, D.Z. and S.N. All authors have read and agreed to the published version of the manuscript.

**Funding:** This research was funded in part by Guangdong Basic and Applied Basic Research Foundation under Grant 2020B1515120060 and in part by Funding for Outstanding Doctoral Dissertation in NUAA under Grant BCXJ23-07.

**Data Availability Statement:** Data are contained within the article.

**Conflicts of Interest:** The authors declare no conflict of interest.

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
