# Peer review of "Studies on High-Resolution Airborne Synthetic Aperture Radar Image Formation with Pseudo-Random Agility of Interpulse Waveform Parameters"

_remotesensing, doi:10.3390/rs16010164_

Round 1
Reviewer 1 Report
Comments and Suggestions for Authors
This paper studied the pseudo-random agility technology of inter-pulse parameters in synthetic aperture radar (SAR) and proposed an imaging processing scheme under the condition of multiple agile parameters. The influences of agility of different parameters were studied, and a practical experiment was conducted to verify the method. The research topic is interesting and of practical value, but there are still some minor issues that need consideration.
1. Abstract: I suppose the paper is mainly focused on the proposed imaging scheme for SAR with multiple agile inter-pulse parameters. I noticed the proposed imaging scheme can deal with multiple agile parameters, not only single ones (in existing literature). I suggest the author should emphasize the advantage in the abstract part.
2. Figure 1: I suggest the authors give the SAR spotlight operation mode in lines 126-127 as the caption in Fig.1. The angle is in the ground plane in Fig.1, which is not the typical squint angle (line 130). Please give it another name.
3. Figure 6: I suggest the authors should give the observation angle differences as described in line 307.
4. Line 321-330: I suggest the authors give a table to describe the variation range, frequency and step-width of different agile parameters in the experiment for clarity.
5. Line 372: Figure 9f gives the well-focused result. I suggest the authors select a strong point-target (if it is feasible to find) in Fig.9f for profile analysis, and evaluate the imaging performance quantitatively.
6. Please give the complete forms of the abbreviations where they first appear in the paper. For example, ‘RF’ in line 94, ‘MEMS based IMU/GPS’ in line 98, ‘PFA’ in line 138 and ‘SNR’ in line 189.
Reviewer 2 Report
Comments and Suggestions for Authors
Improving the accuracy of aerial imaging is a crucial issue addressed in this article. However, I have a few comments outlined below:
Lines 2-4: The subject of the study is very complex and may not be understandable to a 'typical' user without knowledge of InSAR
The introduction is clear and allows for an orientation in the described topic.
Images 4, 5, and 6 don't contribute anything; there's no explanation or demonstration of differences, etc.
What is the difference between Figure 7 and Figure 11? Apart from the captions, the content and scope are exactly the same.
Figure 10 requires an explanation. A dark image with vertical stripes doesn't convey any information, and there's no comparison to the results.
What is the difference between Figure 8 and Figure 12? Besides the captions, the content and scope are exactly the same.
These are not all the comments; I focused only on the graphical part, which emphasizes the scientific nature of the study. There is a lot to improve, add, and explain here.
Reviewer 3 Report
Comments and Suggestions for Authors
In the paper is proposed processing scheme of high resolution SAR image formation which is appropriate for agile interpulse waveform parameters. Simultaneous random agility of four waveform parameters of SAR, namely pulse width, chirp rate polarity, initial phase and PRI, was presented and verified. The complexity and uncertainty of radar waveform were greatly increased, and the anti-interference ability can certainly be improved in principle. Furthemore, the computation load is nearly comparable to that of traditional SAR image formation with constant waveform parameters.
The article is properly designed. It begins with a comprehensive introduction. Chapter 2 takes up the scientific description of the issue under analysis, Chapter 3 makes a description of "4.5kg Microminiature SAR". Chapter 4 is an experimental verification of SAR Image Formation. One would have to consider whether to change the form of presentation/remove some of the illustrations. The last two chapters are discussion and conclusions. Critical remarks - overly extensive citations, e.g. [15,16,22-28], lack of reference to item 19 of the literature, multiple repetition of the phrase "4.5kg Microminiature SAR", the article is in the form of a paper presenting a specific solution, somewhat less scientific. In my opinion, the article deserves to be published after making corrections.
Reviewer 4 Report
Comments and Suggestions for Authors
Manuscript ID: remotesensing-2724689
Title: Studies on High-resolution Airborne SAR Image Formation with Pseudo-random Agility of Interpulse Waveform Parameters
Authors: Zheng Ye, Daiyin Zhu, Shilin Niu, and Jiming Lv
In this work authors aimed to study and verify whether air-borne SAR with agile interpulse waveform parameters can acquire high-quality high-resolution SAR images just like conventional airborne SAR with constant waveform parameters.
The manuscript remotesensing-2724689 is well conceived and written in understandable English. The theory underlying the algorithms used is correctly formulated, the introduction provide sufficient background and include all relevant references, the methods used are comprehensively described by the authors and the results obtained are well detailed. Finally, the conclusions are clear, concise and well highlighted and the cited references are relevant to the research. The only suggestion I can make to the authors in order to further improve the presentation of the paper is to lighten figures 4, 5, 6, 9, 10 and 13. therefore, based on what was expressed previously, I suggest submitting this manuscript to a minor revision process.
